# Impact of Human Disturbances on the Spatial Heterogeneity of Landscape Fragmentation in Qilian Mountain National Park, China

**Baifei Ren** [1], **Keunhyun Park** [2] , **Anil Shrestha** [1], **Jun Yang** [2], **Melissa McHale** [1], **Weilan Bai** [3] **and Guangyu Wang** [1,*]

1  Faculty of Forestry, University of British Columbia, 2424 Main Mall, Vancouver, BC V6T 1Z4, Canada
2  Department of Earth System Science, Tsinghua University, Haidian, Beijing 100084, China
3  China Urban Construction Design and Research Institute, 36 Deshengmen Road, Xicheng District, Beijing 100120, China
*  Correspondence: guangyu.wang@ubc.ca; Tel.: +1-604-822-2681

**Abstract:** Qilian Mountain National Park (QLMNP) is a biodiversity hotspot with great agriculture and tourism resources. With the expansion of human activities, a few areas of the park are experiencing massive landscape transformation, and these areas are also highly ecologically sensitive. Nevertheless, there are substantial differences in the human activities and natural resources of various communities around QLMNP, resulting in heterogeneous landscape degradation. Hence, this study explores the extent and drivers of spatial heterogeneity in landscape fragmentation associated with ecologically vulnerable communities in QLMNP. Multiple ring buffer analysis and geographically weighted regression (GWR) were used to analyze the relationships between landscape fragmentation and variables of human activities and facilities to identify the main factors influencing landscape fragmentation in different regions. The results reveal that human disturbance had a stronger relationship with landscape fragmentation in QLMNP than natural factors do. Among the drivers of landscape fragmentation, the distribution of residential areas and the extension of agricultural land were found to have more significant impacts than tourism. Expansion of cropland had a greater impact on the eastern part of the national park, where overgrazing and farming require further regulation, while tourism affected the landscape fragmentation in the central area of the national park. The point-shaped human disturbance had a larger impact than the linear disturbance. The study findings can be used to formulate a comprehensive plan to determine the extent to which agriculture and tourism should be developed to avoid excessive damage to the ecosystem.

**Keywords:** human disturbance; tourism activity; landscape fragmentation; geographically weighted regression

## 1. Introduction

Landscape fragmentation refers to the dominance of substantially smaller and more isolated patches of the natural habitat [1,2], which represent changes in the landscape pattern and the underlying ecological process [1,3]. The quantification of the spatial heterogeneity of landscape fragmentation can contribute to the understanding of the interaction between the geographic environment and human activities in landscape systems [4].

The destructive scale of human disturbance has been continuously enlarged by extensive human activities and changes in developmental modes [5,6], resulting in changes in the landscape and structural reorganizations. Therefore, it is crucial to reasonably identify the level of landscape fragmentation and the mode of human disturbance, and to analyze the degradation mechanism of the ecosystem caused by intense human disturbance.

## 2. Literature Review

### 2.1. The Effect of Human Disturbance on the Landscape Pattern

Exploring how spatial heterogeneity is influenced by natural and anthropogenic disturbances is an important topic in landscape ecology [7]. Natural habitat fragmentation in protected areas is occasionally attributed to natural disturbances [8,9]. However, usually, the main cause of habitat fragmentation is human intervention [2,10,11], which is a threat to biodiversity conservation [8,12]. With the transformation of economic growth [6], population growth [13,14], and national policies, human activities, including livestock overgrazing [15], illegal mining [6], cropland expansion [16,17], construction of roads [18–20] and built-up areas [21,22], could have negative impacts on natural habitat [16]. These factors drive a distinct but complex pattern of disturbance in natural habitats [23].

Tourism development has become a significant source of human disturbance with an immediate impact on the landscape pattern [24]. The intensity of disturbance varies with the number, scale, and type of scenic spots [25], as well as the volume of tourist influx [26]. Protected areas are surrounded mainly by rural communities. Sometimes, the expansion of villages is accompanied by tourism development in the area around the national park [27], with the land function changing from meeting the living needs of villagers to satisfying the demands of tourists [28]. The changes in land-use patterns in rural areas tend to depend on the proximity to paths, public services, rural centers, settlement areas [29], tourist accommodations, and tourism centers and locations [30].

### 2.2. Landscape Fragmentation in Protected Areas (PA)

Since changes in landscape structure in natural habitats are often caused by natural and human disturbance, quantitative measurements of spatial information of habitat fragmentation are essential for monitoring the conservation of natural resources [31]. Various metrics have been used to evaluate and measure spatial heterogeneity in protected areas [12]. Ianăş et al. [32] used landscape metrics which were divided into 3 categories including landscape composition, shape, and configuration, to show the fragmentation of natural habitat in the Nera Gorges-Beuşniţa National Park. Landscape metrics, such as patch density (PD), landscape shape index (LSI), largest patch index (LPI), landscape division index (DIVISION), modified Simpson's diversity index (MSDI) [33], Euclidean nearest neighbor (ENN) [33,34], mean patch size, and the total number of patches [35], have been used to quantify landscape fragmentation within the national parks and surrounding areas. When assessing the change of natural habitat, most of previous research used PD which can be used to represent the overall heterogeneity and fragmentation of the landscape and the degree of fragmentation of a certain type [32,35]. Rodríguez et al. [36] used Contagion Edge Proportion (CEP) as an indicator of fragmentation of protected areas (PA), and they found that except in Nature Reserves, fragmentation increased in all PA types. Mas [37] showed the spatial heterogeneity of annual rate of deforestation in PA and buffer areas to assess the effectiveness of PAs.

Previous studies in European and America [32–34] mostly found increasingly fragmented natural habitats around protected polygons. There are negative effects of landscape fragmentation within and around PAs and national parks [38], including increasing susceptibility [39], decline in biodiversity [14,31] and increased rates of tree mortality [40]. It is critical to identify fragmentation levels and translate these to establish policies limiting the abuse of protected areas.

### 2.3. Conflict between Environmental Protection and Community Development in QLMNP

The rapid development of agriculture and animal husbandry in the Qilian Mountains poses an environmental threat [41]. Anthropogenic activities, such as illegal mining and the unauthorized construction of facilities [6], have resulted in the increased fragmentation of the natural habitat. Thus, a policy of ecological priority was proposed for the construction of Qilian Mountain National Park (QLMNP). However, environmental protection in these areas has led to severe conflicts among local governments, enterprises, villagers, and other

stakeholders [28,42,43]. The residents mainly rely on agriculture and animal husbandry for their livelihood [16], and the expansion of cultivated land is the major approach for agricultural production and income. But the latest national park master plan prohibits grazing in core protected areas [44], which could have a negative impact on the local economy.

According to the Master Plan of QLMNP, the local community should mainly develop ecotourism and education about nature, provide visitor reception services, identify residential areas for ecological immigrants in the Qilian Mountain, and minimize the interference and impact of human activities on the park environment. Although the QLMNP landscape has great potential for tourism development, its ecology is highly vulnerable [45,46]. In recent years, excessive grazing, the rapid development of tourism, and small hydropower projects have damaged the ecological environment [6].

However, with the establishment of QLMNP and rapid development around QLMNP, there is limited research about communities inside and around QLMP and a lack of studies that have involved the spatial analysis of the human disturbance to the landscape patterns in QLMNP. Thus, this study aims to explore how much the spatial heterogeneity of landscape fragmentation is related to human disturbance in communities in QLMNP. The main research questions are: (1) Is fragmentation happening more heavily in certain areas of the park? (2) If we assume that the fragmentation level of habitat is measurable, are the main drivers of this measurable fragmentation impact within a certain geographical range? (3) Can we attribute these measurable impacts to human disturbance, such as an expansion of cropland and built-up area of human settlement, tourists' activities, transportation infrastructure and so on? Based on the research questions, the result of the paper consists of the following three parts: (1) the patterns of human activities; (2) the spatial heterogeneity of the landscape fragmentation of the natural habitat, (3) the impact of human disturbances on the landscape fragmentation of the natural habitat in each community.

The national park communities suitable for the development of ecotourism and agriculture are then determined based on the inherent requirements of ecological management and control. Under the premise of ecological protection in QLMNP, the more vulnerable communities are identified to determine the development strategies appropriate for different conditions.

## 3. Materials and Methods

### 3.1. Research Area

The QLMNP pilot is located at the junction of Gansu and Qinghai Provinces in China in the northeast region of the Qinghai-Tibet Plateau (Figure 1) and has a total area of 50,200 km$^2$ [46,47]. The Qilian Mountains are a critical ecological shelter in China [45], a vital headwater of the Yellow River Basin, and a priority area for biodiversity conservation in China. It is an agriculture and pasture interlaced zone in northwest China. This is also an ecologically vulnerable area [48,49], as the mountainous area is sensitive to climate change and has poor tolerance to human disturbances [16,50]. QLMNP is located in an area inhabited by ethnic minorities [46] with a rich traditional culture [51]. Some communities with well-developed facilities and ecotourism resources attract visitors and have increased the income of residents [44,52] Although the QLMNP landscape is highly attractive, ecological importance should be taken into consideration in zoning of each community, given that some of these areas are also extremely ecologically sensitive [16,51].

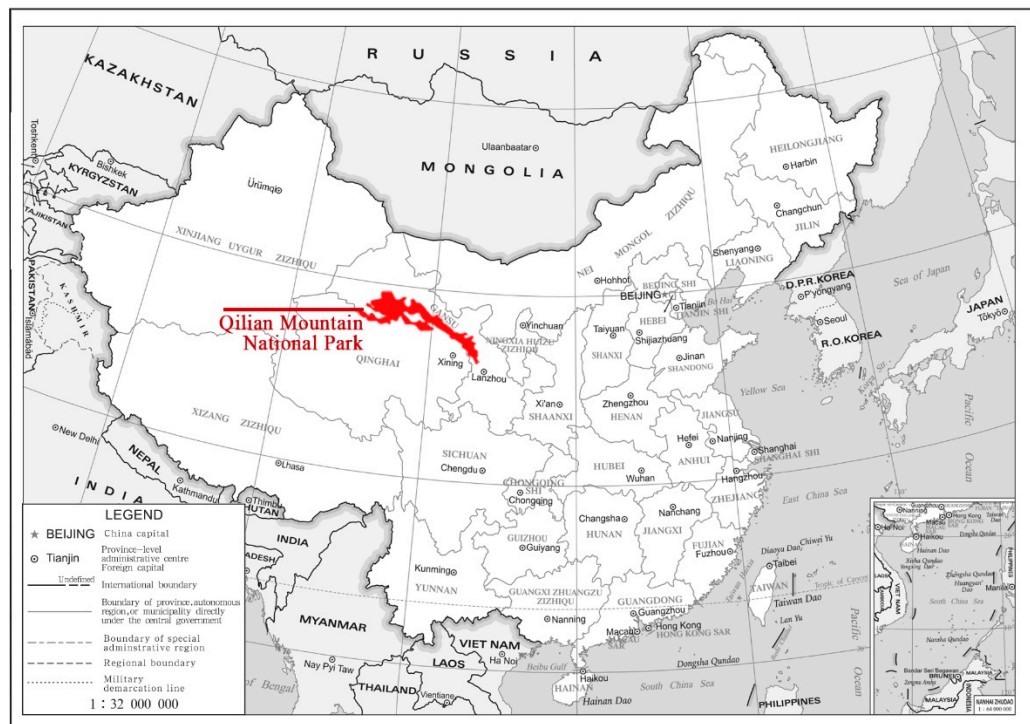

**Figure 1.** The location of the research area (Source: map based on a figure obtained from the Ministry of Natural Resources [53]).

QLMNP is divided into areas of core protection and general control. The core protected area is 27,466 km$^2$, and the general protected area is 22,767.72 km$^2$ [44]. Sixty-three local township communities in the northwest region of QLMNP were selected as the research area, including gateway communities and communities passing through the general protected area (Figure 2).

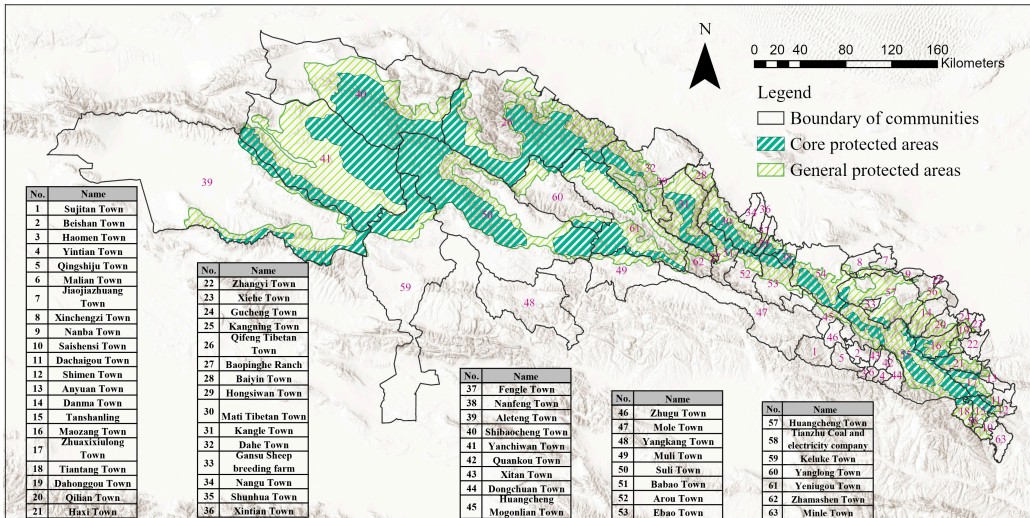

**Figure 2.** The control zone and the boundary of the township communities of QLMNP (Source: redrawn by author in ArcGIS based on data from the Resource and Environment Data Cloud Platform).

### 3.2. Overall Research Design

Figure 3 presents the research workflow. First, the land cover/land use (LCLU) data were used to calculate the landscape fragmentation indexes, including the patch density (PD) and splitting index (SPLIT), with Fragstats 4.2 software. The "Zonal statistics as table" tool in ArcGIS Pro 2.8 was used to calculate the mean values of possible human disturbance factors in each community. Based on a previous study, nine factors of human disturbance and three factors of the natural background were selected as independent variables (Table 1). We would like to choose human settlements and residential areas, transport and travel demand, human infrastructure, and agricultural development as possible anthropogenic drivers of habitat fragmentation. Based on previous study, hydropower stations and mining sites are also important factors that may have influence on landscape fragmentation and natural environment [6]. But because these data were temporarily unavailable to us, we only select the indicators in Table 1 for analysis. Since this paper mainly discussed the impact of anthropogenic activities on habitat fragmentation, natural factors were not discussed in depth. Then, the human activities in each community of the national park were analyzed. The geographically weighted regression (GWR) model was used to analyze the impacts of various human disturbances on the landscape fragmentation indexes in the communities. Additionally, a multi-ring buffer analysis of the spatial heterogeneity of the landscape variation coefficient (LVC), PD, and SPLIT was performed in adjacent buffer distances under different types of human disturbance with typical sources of disturbance at the center.

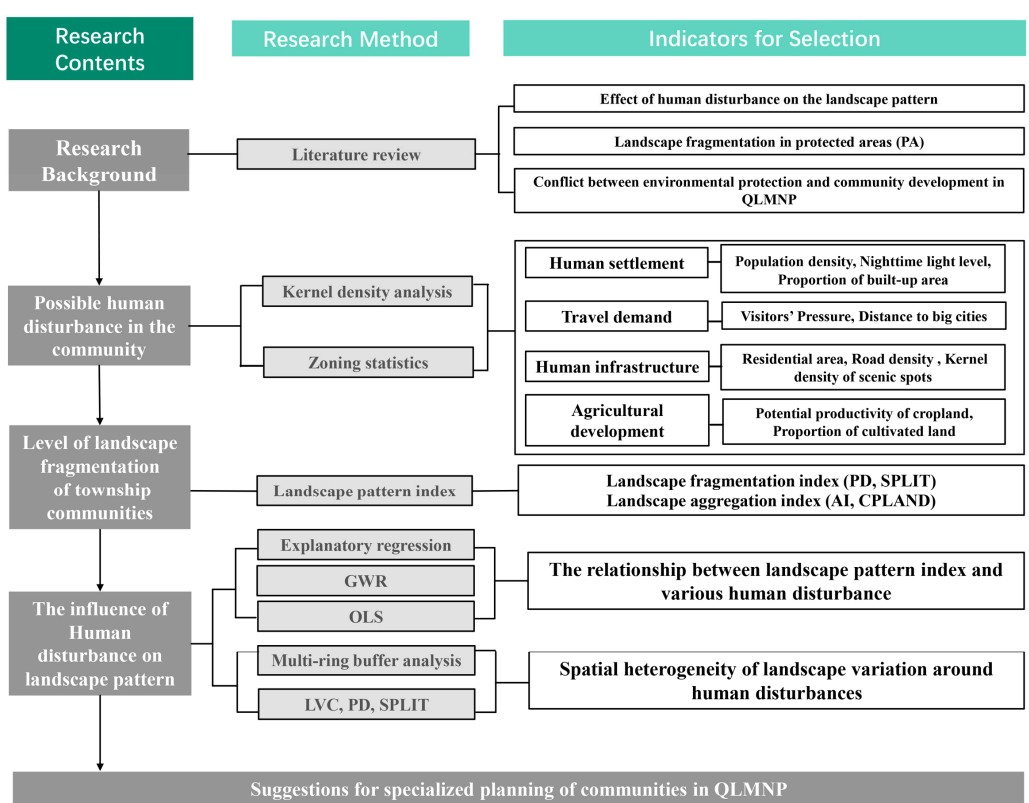

**Figure 3.** The research workflow.

**Table 1.** The indicators of the impact of human activities on the landscape pattern.

| Category | Indicators | References Using Indicators of the Human Impact on the Landscape Pattern |
|---|---|---|
| Human settlements and residential areas | Nighttime light level | Levin et al. [54], Huang et al. [55] |
| | Population density | Wittemyer et al. [13], Burgess et al. [14], Nagendra et al. [11] |
| | Proportion of the built-up area | Liu et al. [21], Zeng et al. [22] |
| Transport and travel demand | Visitor pressure | Orsi et al. [26] |
| | Distance to source markets | Rodríguez et al. [12] |
| Human infrastructure | Density of scenic spots | Xiang et al., 2019 [25] |
| | Road density | Cai et al. [18], Hawbaker et al. [19], Sánchez-fernández et al. [20] |
| Agricultural development | Proportion of cultivated land | Qian et al. [16], Mottet et al. [17] |
| | Potential cropland productivity | Wang et al. [56] |
| Natural background | Annual mean temperature | Qian et al. [16] |
| | Annual mean precipitation | Qian et al. [16] |
| | Terrain niche index | Gong et al. [50], Pei et al. [57] |

*3.3. Research Methodology*

3.3.1. Geographically Weighted Regression (GWR)

The same variable has different fitting effects in different regions. In this study, GWR was used to analyze the spatial heterogeneity of the impacts of human disturbances and socioeconomic factors on the landscape fragmentation of the natural habitat in communities of QLMNP. First, the data were standardized by normalization by using the range method, as follows.

$$x_{ij}^* = \frac{\max(x_i) - x_{ij}}{\max(x_i) - \min(x_i)} \qquad (1)$$

The GWR model was constructed in ArcGIS Pro 2.8. The basic GWR relationship is represented as follows:

$$y_i = \beta_0(u_j, v_j) + \sum_{i=1}^{p} \beta_i(u_j, v_j) x_{ij} + \varepsilon \qquad (2)$$

In Equation (2), $(u_j, v_j)$ represents the coordinates at location $j$ and $\beta_0(u_j, v_j)$ is the intercept coefficient at location $j$, which is the local regression coefficient for independent variable $x_i$ at location $j$ [58].

3.3.2. Indicators of Landscape Fragmentation

The land-use types, including residential areas, mining sites, roads and other artificial surface are combined into built-up land. The land mainly planted with crops was identified as cultivated land, while forests, shrubland, grassland, water, snow, and glaciers were combined as the natural habitat [59,60]. The moving window method was used to calculate the splitting index (SPLIT), patch density (PD), aggregation index (AI), and core area percentage of landscape (CPLAND) in Fragstats 4.2. SPLIT and PD are always used to show the fragmentation level of natural habitat [33,61]; AI and CPLAND are used to assess the agglomeration of natural habitat [62]. And landscape variation coefficient (LVC) is calculated in ArcGIS Pro 2.8. The formula of LVC is:

$$LVC_i = \sum_{n=1}^{\infty} |a_i - a'_i| \qquad (3)$$

$LVC_i$ is the proportion of the coefficient of variation of landscape type $i$, and $a_i$ and $a'_i$ respectively denote the proportions of the areas of two adjacent buffer zones for comparison.

A distance attenuation curve of typical man-made interference intensity can be established according to the landscape variation coefficient (LVC) under different buffer distances [25].

Additionally, because the geographic area of community in the central and western Qilian Mountains is relatively large, a 20 km × 20 km fishnet was created in ArcGIS to better represent the landscape pattern distribution of large-scale communities+ in the western and central parts of QLMNP. The land-use data were segmented into 1175 plots measuring 20 km × 20 km with fishnets, and the landscape pattern index of the natural habitat was calculated for each plot. According to the natural breakpoint method, the landscape pattern index was divided into eight categories.

### 3.3.3. Disturbance Intensity Based on the Landscape Fragmentation Indexes

When evaluating the impacts of different types of human disturbance, the multiple ring buffer analysis method [25,35,63] was used to analyze the variations of the landscape fragmentation indexes in adjacent buffer zones under different types of human disturbance with typical disturbance sources at the center.

Four main types of human interferences were identified in this study: (1) linear interference that spreads to both sides with main roads as the source; (2) point-shaped interference spreading to surrounding areas with residential areas (location of residents' committees) as the source; (3) point-shaped interference spreading to the surrounding areas with important scenic spots as the source; (4) tourists' preferred travel routes extracted according to the kernel density of track point data of the tourists. A control group was also established by using 68 random points in the natural habitat as sources of natural interference for comparative analysis, which can be interpreted as control variables.

### 3.4. Data Sources

The proportions of the built-up area and cultivated land were obtained from LCLU data with a 30 m resolution (Table 2). The LCLU data used were derived from GlobeLand30. Road and traffic network data, including data on highways, railways, and provincial, county, and national roads, were downloaded from the Resource and Environment Data Cloud Platform established by the Chinese Academy of Sciences Resources and Environment Center. The data on top-grade scenic spots and national forest parks in Qinghai and Gansu Provinces were mainly obtained from the website of the Tourism Bureau of the Provincial Department of Culture. The geocoding of AutoNavi API was used to identify the latitude and longitude of each tourist attraction and residents' committee, including the committees of nomadic people and villagers.

**Table 2.** The data source and calculation methods of indicators used in the study.

| Element | Indicators | Method/Equation | Data Source |
|---|---|---|---|
| Human settlements and residential areas | Nighttime light level | Zonal statistics | Chen et al. [64] |
| | Population density | Zonal statistics | WorldPop |
| | Proportion of the built-up area | Summary statistics | LCLU data. Source: Global 30 [65] |
| Transport and travel demand | Visitor pressure | Kernel density of tourist routes $f(s) = \sum_{i=1}^{\infty} \frac{1}{h^2} k\left(\frac{s-c_i}{h}\right)$ | 711 trajectories, (GPX version) from Xingzhe; 677 trajectories (GPX version) from Foooooot; 29 trajectories (GPX version) from Wikiloc; Total 4,390,707 data points |
| | Distance to large cities | Euclidean distance | Resource and Environment Data Cloud Platform |

**Table 2.** *Cont.*

| Element | Indicators | Method/Equation | Data Source |
|---|---|---|---|
| Human infrastructure | Density of scenic spots | Kernel density $f(s) = \sum_{i=1}^{\infty} \frac{1}{h^2} k\left(\frac{s - c_i}{h}\right)$ | Documents of the Departments of Culture and Tourism of Qinghai and Gansu Provinces |
| | Road density | Summary statistics | Resource and Environment Data Cloud Platform |
| Agricultural development | Proportion of cultivated land in each community | Summary statistics | Land-use/cover data |
| | Mean value of the potential productivity of cropland | Zonal statistics | Resource and Environment Data Cloud Platform |
| Natural background | Annual mean temperature | Zonal statistics | Resource and Environment Data Cloud Platform |
| | Annual mean precipitation | Zonal statistics | Resource and Environment Data Cloud Platform |
| | Terrain niche index [57] | Raster calculator: $T = log\left[\left(\frac{E}{\bar{E}} + 1\right) \times \left(\frac{S}{\bar{S}} + 1\right)\right]$; Zonal statistics | Resource and Environment Data Cloud Platform |

Tourist GPS trajectory data were obtained from three websites of outdoor activities, namely Foooooot, Xingzhe, and Wikiloc. The users of the Foooooot and Xingzhe software programs are mainly Chinese tourists, including those involved in outdoor biking activities in Xingzhe [66,67]. The majority of Wikiloc users are non-Chinese users. The trajectory data were preprocessed to exclude data outside the study area and duplicate data. Finally, a total of 1417 pieces of trajectory data, including 4,390,707 points, were used for further data analysis. The kernel density was calculated based on the trajectory data to show tourist routes and the distribution of visitor pressure. The sources of other data, such as the nighttime light level and the potential productivity of cropland, are presented in Table 2.

## 4. Results

### 4.1. Patterns of Human Activities and Other Drivers

4.1.1. Spatial Distribution of Transport and Travel Demand

(1) Kernel density of tourist routes: The tourist routes are concentrated in the middle of QLMNP. Tourist resources such as ethnic minority towns, grasslands, and snowy mountains exist along the route. The tourist routes differ from the existing traffic lines. Many tourist routes pass through the core protected area of the national park (Figure 4a–c).

(2) Distance to cities: The eastern edge of QLMNP is closer to provincial capitals and prefecture-level downtown areas, which are closer to the source of visitors, while the western communities of QLMNP are far from the sources (Figure 4).

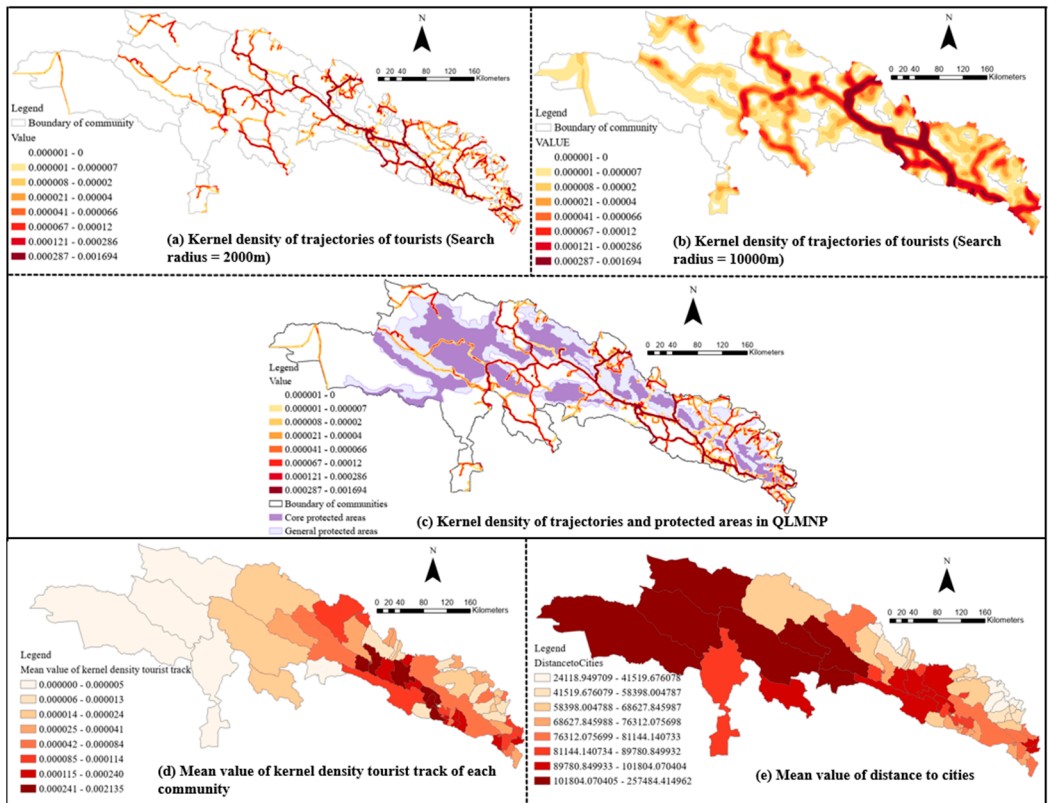

**Figure 4.** The spatial distributions of transportation and travel demand: (**a**) the kernel density of tourist routes (Search radius = 2000 m); (**b**) the kernel density of tourist routes (search radius = 10,000 m); (**c**) the kernel density of trajectories and protected areas in QLMNP; (**d**) the mean value of the kernel density of the tourist route of each community; (**e**) the mean value of the distance to large cities.

### 4.1.2. Spatial Distributions of Human Settlements and Residential Areas

(1) Nighttime light level: The nighttime light has always been used to measure human activity [54]. The nighttime data shows that the eastern edge of the national park has the high brightness, reflecting increased human activity and socioeconomic attributes. The brightness values of the urban areas in the central and southern Qilian Mountains are also higher than those at the western edge (Figure 5a).

(2) Population density: The densely populated communities are mainly concentrated on the eastern edge of QLMNP (Figure 5c).

(3) Proportion of the built-up area: The communities with a higher proportion of the built-up area are mainly concentrated on the southern and eastern edges of QLMNP, while communities with a lower proportion of the built-up area are mainly located in the western portion of QLMNP (Figure 5b).

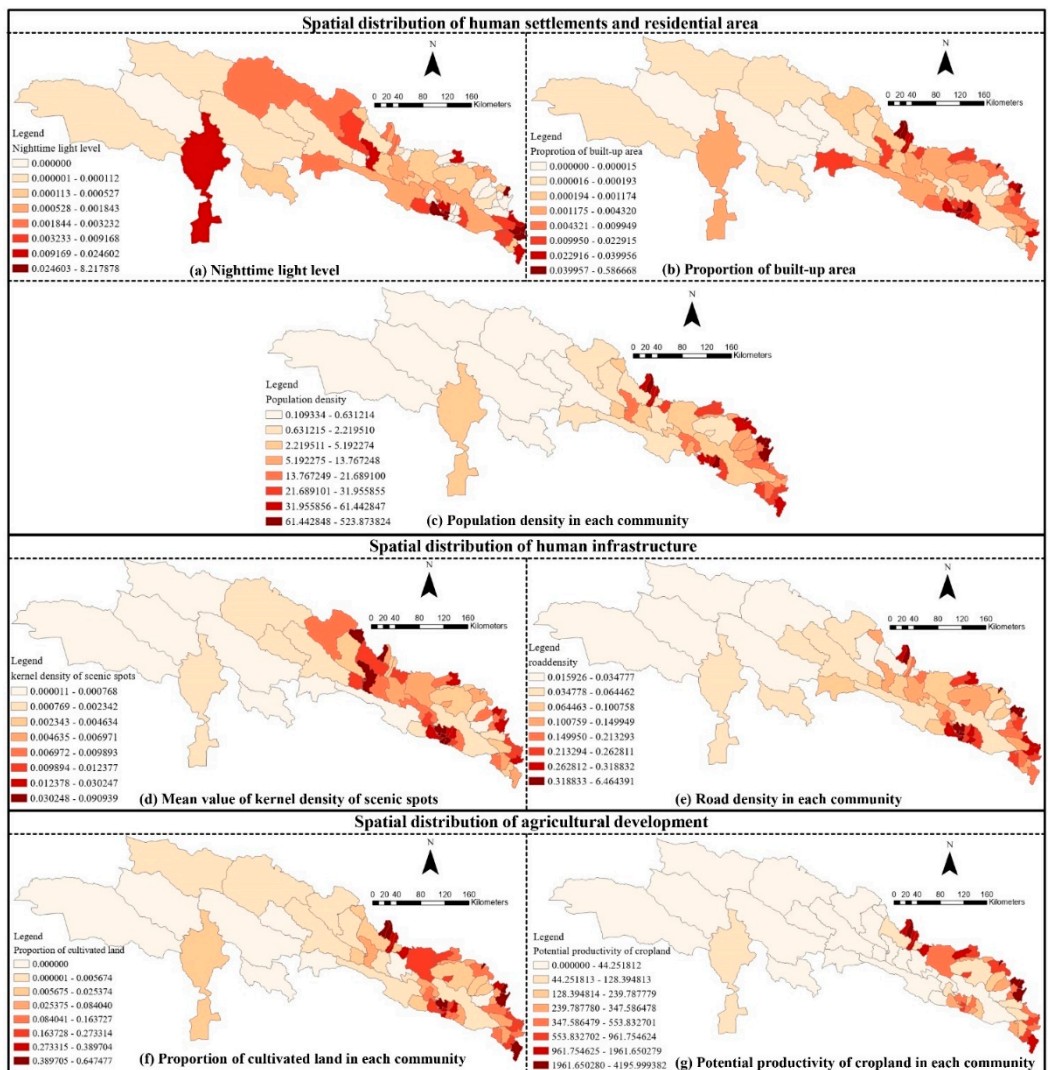

**Figure 5.** The spatial distributions of human settlements, infrastructure, and agricultural development in communities of QLMNP: (**a**) the nighttime light level; (**b**) the proportion of the built-up area; (**c**) the population density in each community; (**d**) the mean value of the kernel density of scenic spots; (**e**) the road density in each community; (**f**) the proportion of cultivated land in each community; (**g**) the potential productivity of cropland in each community.

4.1.3. Spatial Distributions of Human Infrastructure

(1) Density of scenic spots: The density of scenic spots and tourist attractions is higher on the northern slope of the eastern part of the Qilian Mountains than in other communities (Figure 5d).

(2) Road network density: Communities with higher road network density are mainly concentrated on the eastern edge of QLMNP (Figure 5e).

### 4.1.4. Spatial Distribution of Agricultural Development

Communities with higher potential cropland productivity and cultivated land are mainly concentrated on the eastern edge of QLMNP, while the potential productivity of cropland and the proportion of cultivated land in the communities located in the western portion of the Qilian Mountains, such as Yanchiwan Town, Aleteng Town, and Suli Town, are lower (Figure 5f,g).

### 4.1.5. Spatial Distributions of Natural Factors

(1) Annual mean temperature: Areas with a high annual mean temperature are mainly concentrated in the periphery of QLMNP (Figure 6a).

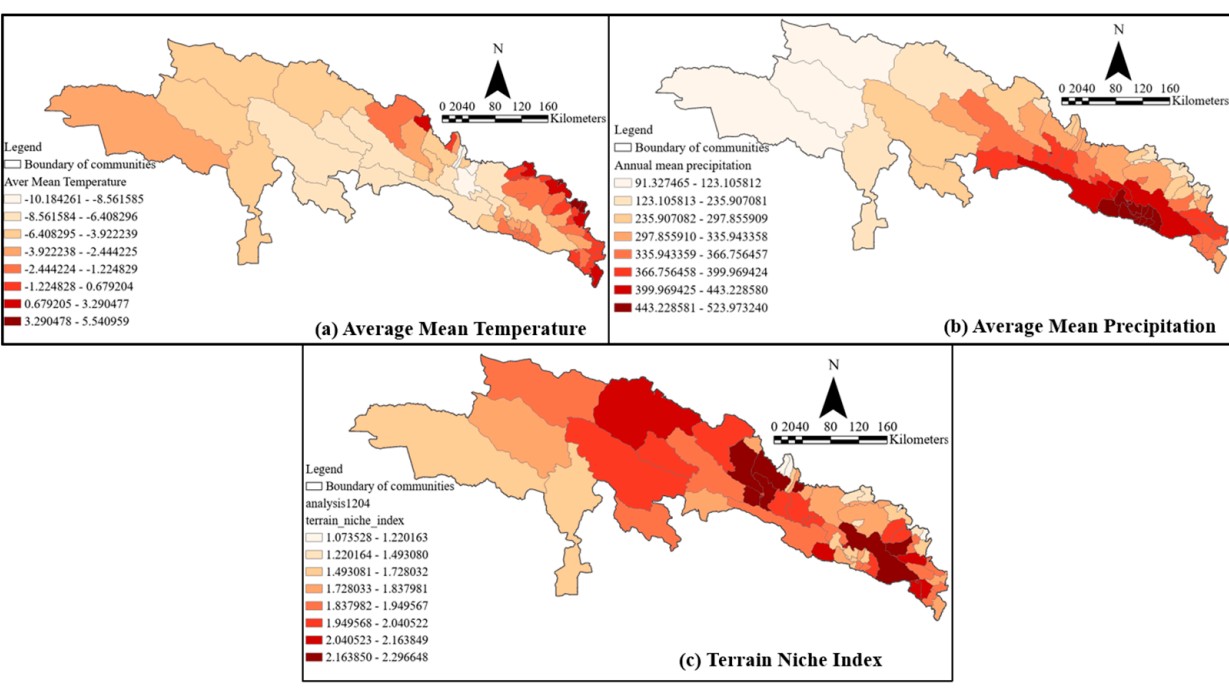

**Figure 6.** The spatial distributions of natural factors: (**a**) the average mean temperature; (**b**) the average mean precipitation; (**c**) the terrain niche index.

(2) Annual mean precipitation: Areas with a high average mean precipitation are mainly located on the southern slope of the Qilian Mountains (Figure 6b).
(3) Terrain niche index: The terrain niche index is a combination of slope and elevation [57]. Communities with a high terrain niche index are mainly concentrated in the central area of QLMNP (Figure 6c).

### 4.2. Patterns of Landscape Fragmentation

The communities in natural habitats with higher values of the landscape fragmentation indexes (SPLIT and PD) are mainly concentrated in the eastern part of QLMNP (Figure 7). Areas with high landscape fragmentation indexes are mainly concentrated at the edge of QLMNP and the communities in the peripheral area of QLMNP. The areas of communities in the west are relatively large, such as that of Keluke Town, where the fragmentation index of the natural habitat is low on the community-level map (Figure 7). In comparison, the areas with low levels of the landscape fragmentation indexes and a high level of the landscape aggregation index are mainly located in the fringe area of Keluke town close to the city of Delingha (Figure 7).

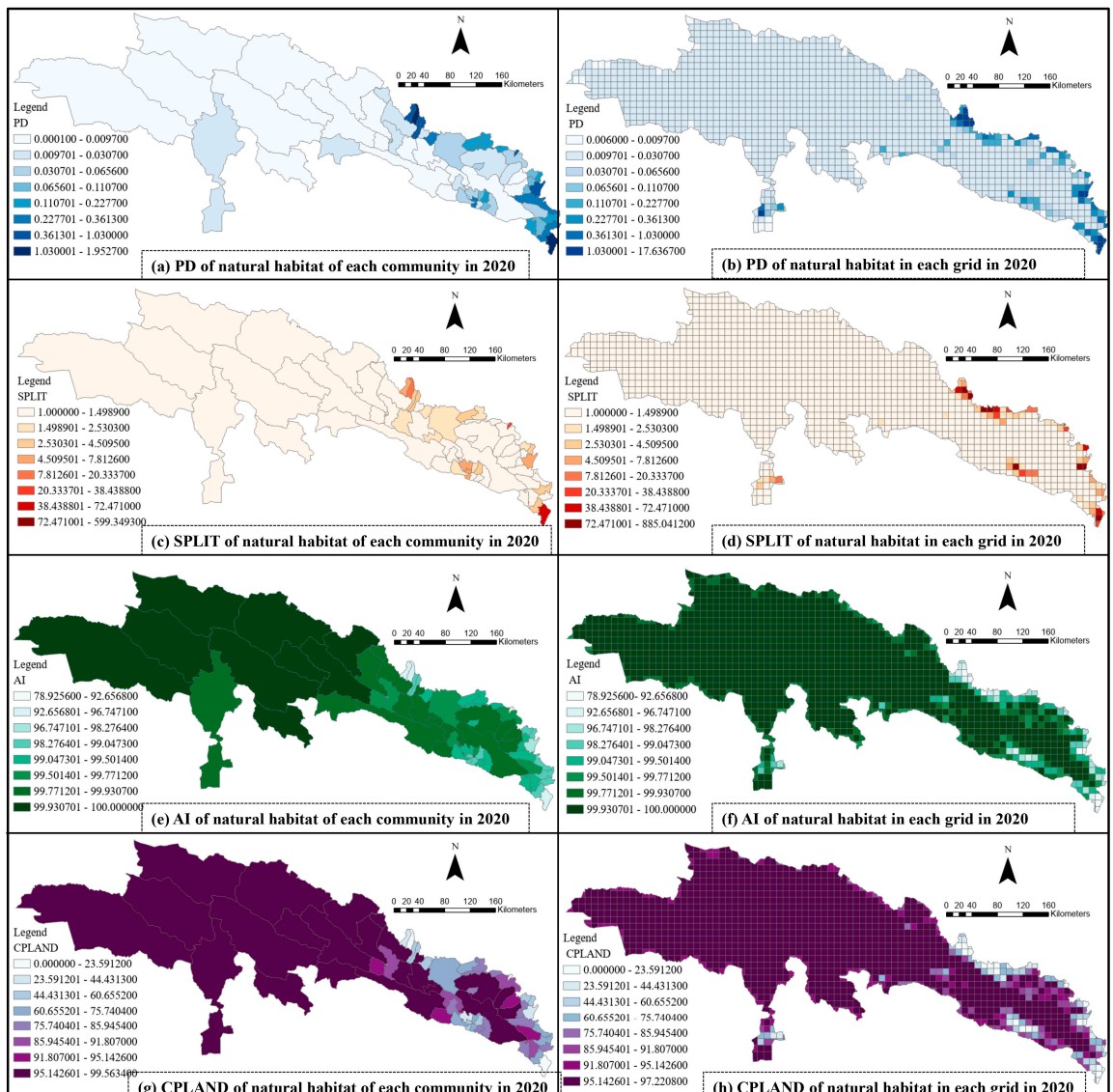

**Figure 7.** The spatial distributions of agricultural development in each community: (**a**) the PD of the natural habitat of each community and (**b**) in each grid in 2020; (**c**) the SPLIT of the natural habitat of each community and (**d**) in each grid in 2020; (**e**) the AI of the natural habitat of each community and (**f**) in each grid in 2020; (**g**) the CPLAND of the natural habitat of each community and (**h**) in each grid in 2020.

Since the QLMNP is at the junction of the two provinces, and the southern part of the Qilian Mountains is mainly in Qinghai Province, while the northern part of the Qilian Mountains is in Gansu Province, two colors are used to represent the landscape pattern index of the natural habitats of the communities in the two provinces. In QLMNP, the landscape fragmentation indexes of the natural habitat of the communities in Qinghai are relatively low and stable (Figure 8a,b), while the landscape aggregation index of the habitat in Qinghai is relatively high. The landscape fragmentation indexes of communities in Gansu Province are relatively high with large fluctuations, especially in Hongsiwan Town (Figure 8c,d), while the landscape aggregation indexes in Gansu are relatively low.

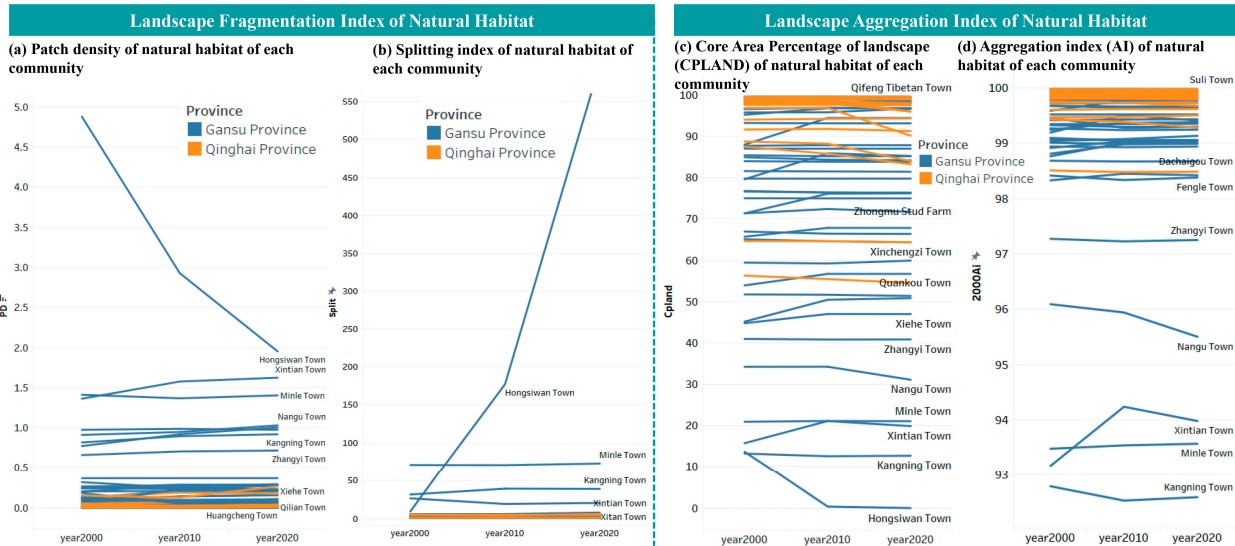

**Figure 8.** The landscape fragmentation indexes and landscape aggregation index of the natural habitat of each community in QLMNP: (**a**) the PD of the natural habitat of each community; (**b**) the SPLIT of the natural habitat of each community; (**c**) the CPLAND of the natural habitat of each community; (**d**) the AI of the natural habitat of each community.

### 4.3. Association of Human Activities to Landscape Fragmentation

#### 4.3.1. Results of the GWR Model of the Spatial Relationships between Landscape Fragmentation and Human Disturbance in Communities

The PD and SPLIT represent the landscape fragmentation indexes. An explanatory regression of the landscape fragmentation indexes (PD, SPLIT) and 12 independent indexes were performed. After removing multicollinear variables, five variables were selected for GWR analysis according to the AICc and adjusted $R^2$ values of the explanatory regression (Appendix A Tables A1 and A2). The independent variable of the proportion of the built-up area was removed from the GWR model, as it had a negative coefficient due to multicollinearity (Appendix A Tables A1 and A2), while it exhibited a positive coefficient in the highest adjusted $R^2$ explanatory regression with one independent variable (Appendix A Tables A3 and A4).

The factor of road density has the largest coefficient and is significant in both two OLS regression models (PD and SPLIT) (Tables 3 and 4). The travel demand factors, including the distance to large cities and visitor pressure, were not significant in the regression models, and exhibited no relationship with the PD or SPLIT. Human factors were found to have a greater impact on the PD and SPLIT, especially the variables of proportion of cultivated land and road density, which are significant in both PD and SPLIT models; while natural factors, such as the annual mean precipitation and annual mean temperature, may have a lesser impact (Tables 3 and 4).

**Table 3.** The results of the OLS model including the PD and 12 selected independent variables.

| Variable | Coefficient | StdError | t-Statistic | Probability | Robust SE | Robust t | Robust Pr |
|---|---|---|---|---|---|---|---|
| Intercept | 0.0179 | 0.0722 | 0.2484 | 0.8048 | 0.0562 | 0.3194 | 0.7507 |
| Potential productivity of cropland | 0.0259 | 0.1244 | 0.2079 | 0.8361 | 0.1210 | 0.2138 | 0.8316 |
| Terrain niche index | 0.1309 | 0.0717 | 1.8263 | 0.0738 | 0.0659 | 1.9877 | 0.0523 |
| Kernel density of scenic spots | −0.0862 | 0.0910 | −0.9473 | 0.3481 | 0.0681 | −1.2647 | 0.2118 |
| Proportion of cultivated land | 0.2737 | 0.0732 | 3.7372 | 0.000480 * | 0.1157 | 2.3657 | 0.021913 * |
| Proportion of the built-up area | 0.2705 | 0.6601 | 0.4098 | 0.6837 | 0.7596 | 0.3561 | 0.7233 |
| Road density | 2.7354 | 0.2699 | 10.1344 | <0.000001 * | 0.4263 | 6.4168 | <0.000001 * |
| Distance to large cities | −0.0909 | 0.0671 | −1.3555 | 0.1813 | 0.0508 | −1.7890 | 0.0797 |
| Annual mean temperature | −0.0857 | 0.0503 | −1.7039 | 0.0946 | 0.0304 | −2.8162 | 0.006939 * |
| Annual mean precipitation | −0.1245 | 0.0522 | −2.3847 | 0.020927 * | 0.0390 | −3.1964 | 0.002414 * |
| Visitor pressure | −0.0759 | 0.2259 | −0.3362 | 0.7381 | 0.1482 | −0.5125 | 0.6106 |
| Nighttime light level | −1.0701 | 0.5778 | −1.8520 | 0.0699 | 0.6543 | −1.6353 | 0.1083 |
| Population density | −0.8283 | 0.3840 | −2.1572 | 0.035818 * | 0.4401 | −1.8822 | 0.0656 |

Note: * indicates $p < 0.05$.

**Table 4.** The results of the OLS model including the SPLIT and 12 selected independent variables.

| Variable | Coefficient | StdError | t-Statistic | Probability | Robust SE | Robust t | Robust Pr |
|---|---|---|---|---|---|---|---|
| Intercept | −0.0268 | 0.0131 | −2.0377 | 0.0469 * | 0.0189 | −1.4179 | 0.1624 |
| Potential productivity of cropland | −0.0056 | 0.0226 | −0.2488 | 0.8046 | 0.0204 | −0.2761 | 0.7836 |
| Terrain niche index | 0.0197 | 0.0130 | 1.5087 | 0.1377 | 0.0170 | 1.1565 | 0.2530 |
| Scenic spots | −0.0248 | 0.0165 | −1.5020 | 0.1394 | 0.0137 | −1.8102 | 0.0763 |
| Proportion of cultivated land | 0.0365 | 0.0133 | 2.7365 | 0.0086 * | 0.0235 | 1.5543 | 0.1264 |
| Proportion of the built-up area | −0.1479 | 0.1201 | −1.2315 | 0.2239 | 0.1084 | −1.3647 | 0.1785 |
| Road density | 0.1942 | 0.0491 | 3.9546 | 0.0002 * | 0.0632 | 3.0715 | 0.0034 * |
| Distance to large cities | 0.0159 | 0.0122 | 1.3066 | 0.1973 | 0.0117 | 1.3586 | 0.1804 |
| Annual mean temperature | 0.0186 | 0.0092 | 2.0345 | 0.0472 * | 0.0133 | 1.4008 | 0.1674 |
| Annual mean precipitation | −0.0078 | 0.0095 | −0.8180 | 0.4172 | 0.0083 | −0.9405 | 0.3515 |
| Visitor pressure | 0.0338 | 0.0411 | 0.8230 | 0.4144 | 0.0275 | 1.2294 | 0.2247 |
| Nighttime light level | 0.9488 | 0.1051 | 9.0275 | <0.0001 * | 0.1002 | 9.4687 | <0.000001 * |
| Population density | −0.0163 | 0.0698 | −0.2333 | 0.8165 | 0.0623 | −0.2618 | 0.7946 |

Note: * indicates $p < 0.05$.

Based on AICc and adjusted R$^2$ values of results of OLS and GWR models (Appendix A Tables A5 and A6), GWR model can be used for explanation. Based on GWR model, the effect of spatial heterogeneity on the coefficient of human disturbance was reflected in the difference between the eastern and western regions. In contrast, the influences of spatial

heterogeneity on the coefficients of natural variables were mainly reflected in the difference between the northern and southern regions. The road density was found to have a smaller impact on the landscape fragmentation of the natural habitat in the western area of the Qilian Mountains than in the eastern area (Figure 9c). The proportion of cultivated land was found to have a smaller impact on the PD of the eastern and central QLMNP, while the road density and built-up area may have larger impact on fragmentation than the proportion of cultivated land in eastern QLMNP. The higher the number of scenic spots in the community, the lower the level of landscape fragmentation, which is consistent with the important scenic spots in the Qilian Mountains including mainly snow-capped mountains, such as the Danxia geomorphic zone and other complete natural landscapes. Moreover, the closer the distance to large cities, the higher the levels of the landscape fragmentation index. In the central and eastern QLMNP, the distance to large cities has a far greater impact on the landscape pattern than in the western region (Figure 9d).

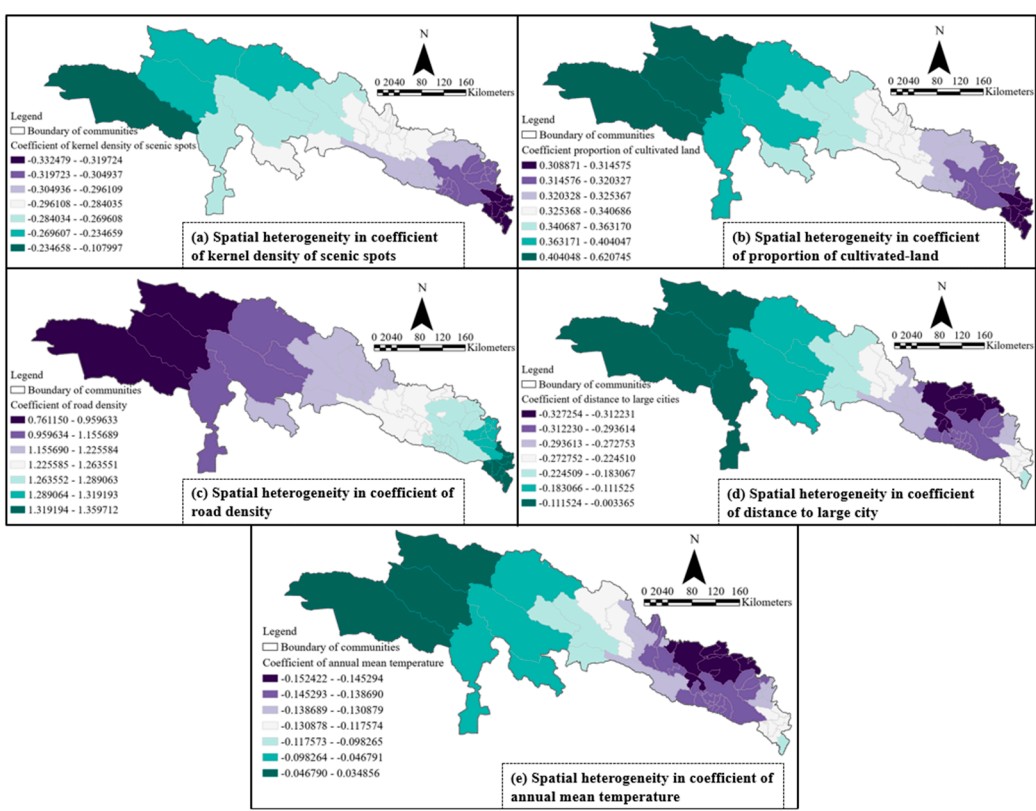

**Figure 9.** The spatial heterogeneity in the coefficients of independent variables based on the PD as the dependent variable.

Human disturbances, such as the proportion of cultivated land, road density, and night-time light level, have a significant impact on the splitting index (SPLIT). The impacts of the road density and cultivated land on the eastern part of the Qilian Mountains were greater than those in the western area (Figure 10). The coefficient of the kernel density of scenic spots in GWR can vary from negative to positive in different communities from east to west. The value of the kernel density of scenic spots tends to be negative; this is likely because, in the eastern region, scenic spots are often located in areas with undisturbed natural habitats, while the western region is exposed to harsh natural conditions over vast uninhabited areas. The locations of scenic spots are often close to the built-up land of villages and towns, and the natural habitats in these areas are characterized by a higher level of fragmentation.

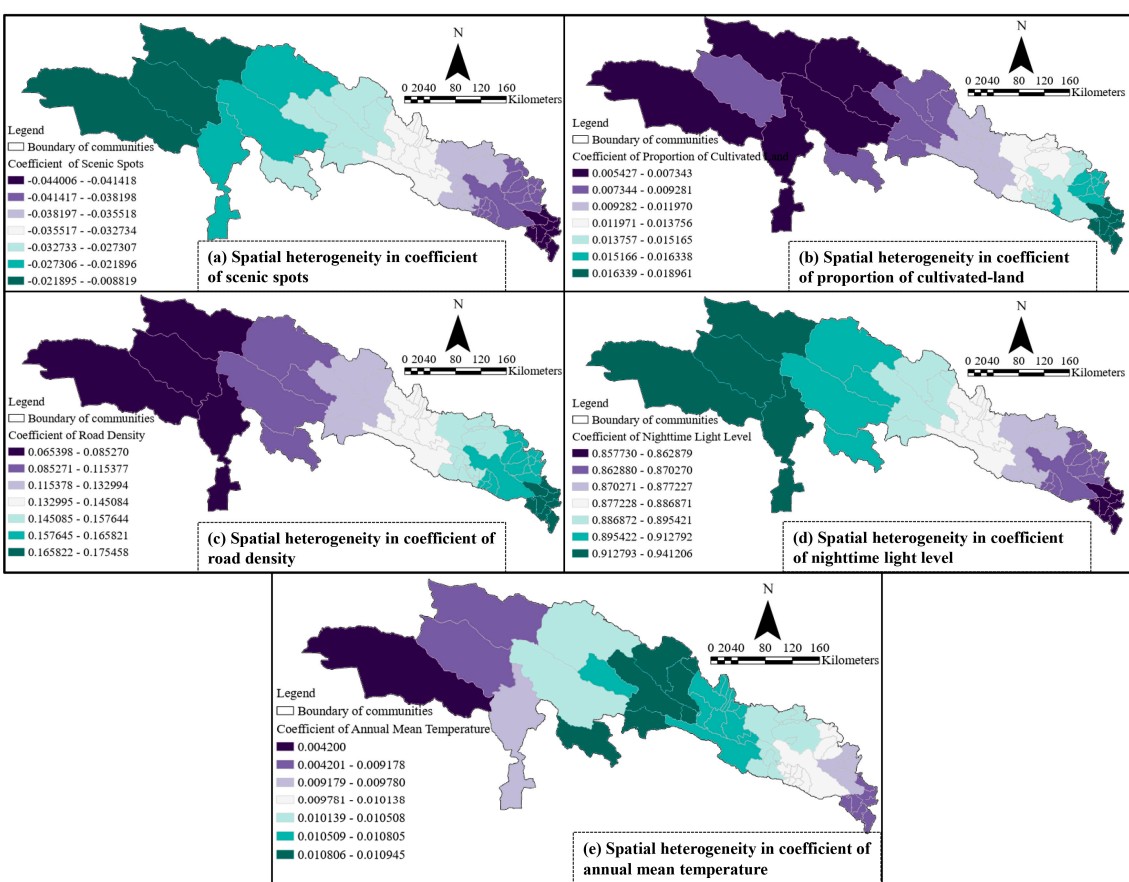

**Figure 10.** The spatial heterogeneity in the coefficients of the road density, nighttime light level, and average annual temperature with the SPLIT as the dependent variable.

### 4.3.2. Multi-Ring Buffer Analysis of Landscape Fragmentation around Human Disturbances

(1)    Spatial variations caused by different sources of disturbance

Within the short-distance buffer zone, the landscape fragmentation indexes tend to be higher, and the overall coefficient of variation tends to show large fluctuations. When the distance to the human disturbance source reaches a certain distance, the level of the landscape fragmentation indexes is significantly reduced. The value of landscape fragmentation index no longer decreases significantly, but instead remains stable and similar to the fragmentation index of random points in natural habitats. When the distance to the human disturbance source reaches a certain threshold, the values of the landscape fragmentation metrics become low and stable, fluctuating within a narrow range, indicating that the impact of human activities on the landscape pattern is so small that it could be ignored. Therefore, the approximate range of influence of a specific type of human disturbance can be delineated based on this distance (Figure 11a,b).

Compared with the random points in the natural background, human interference has led to a significant increase in the LVC of the surrounding areas. The intensity of residential area interference in the surrounding landscape is larger than that of any other interference. The LVC is large in the buffer zone of the settlement within 2400 m. The overall LVC tends to be stable and unified when the distance from the interference source reaches 3000 m (Figure 11c).

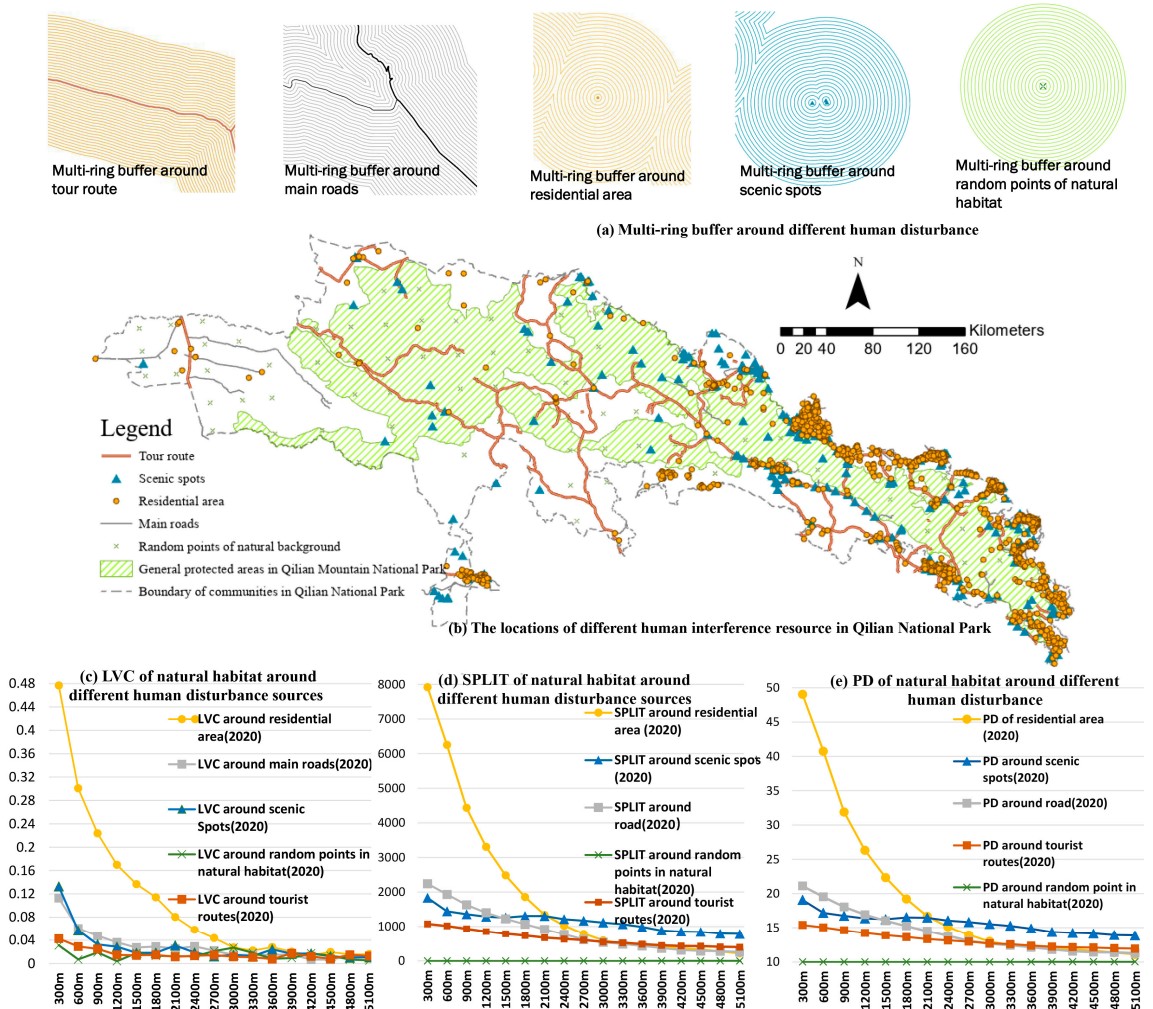

**Figure 11.** The results of landscape fragmentation analysis (LVC, PD, SPLIT) of the multi-ring buffer zones around different human resources.

(2) Changes in the impacts of various sources of human disturbance on landscape fragmentation between two periods (2000 and 2020)

The results of 2000 and 2020 are similar in some respects, e.g., the intensity of the interference of residential areas in the surrounding landscape is larger than that of any other disturbance, including roads, scenic spots, and tourist routes (Figure 12). Compared with 2000, the LVC, PD, and SPLIT of the areas around the residential areas and scenic spots in 2020 were found to exhibit the most significant increases as compared to any other disturbance source. The LVC, PD, and SPLIT of the areas around the scenic spots in 2020 exhibited larger increases than those of roads and tourist routes, but the increment was smaller than that of residential areas (Figure 12).

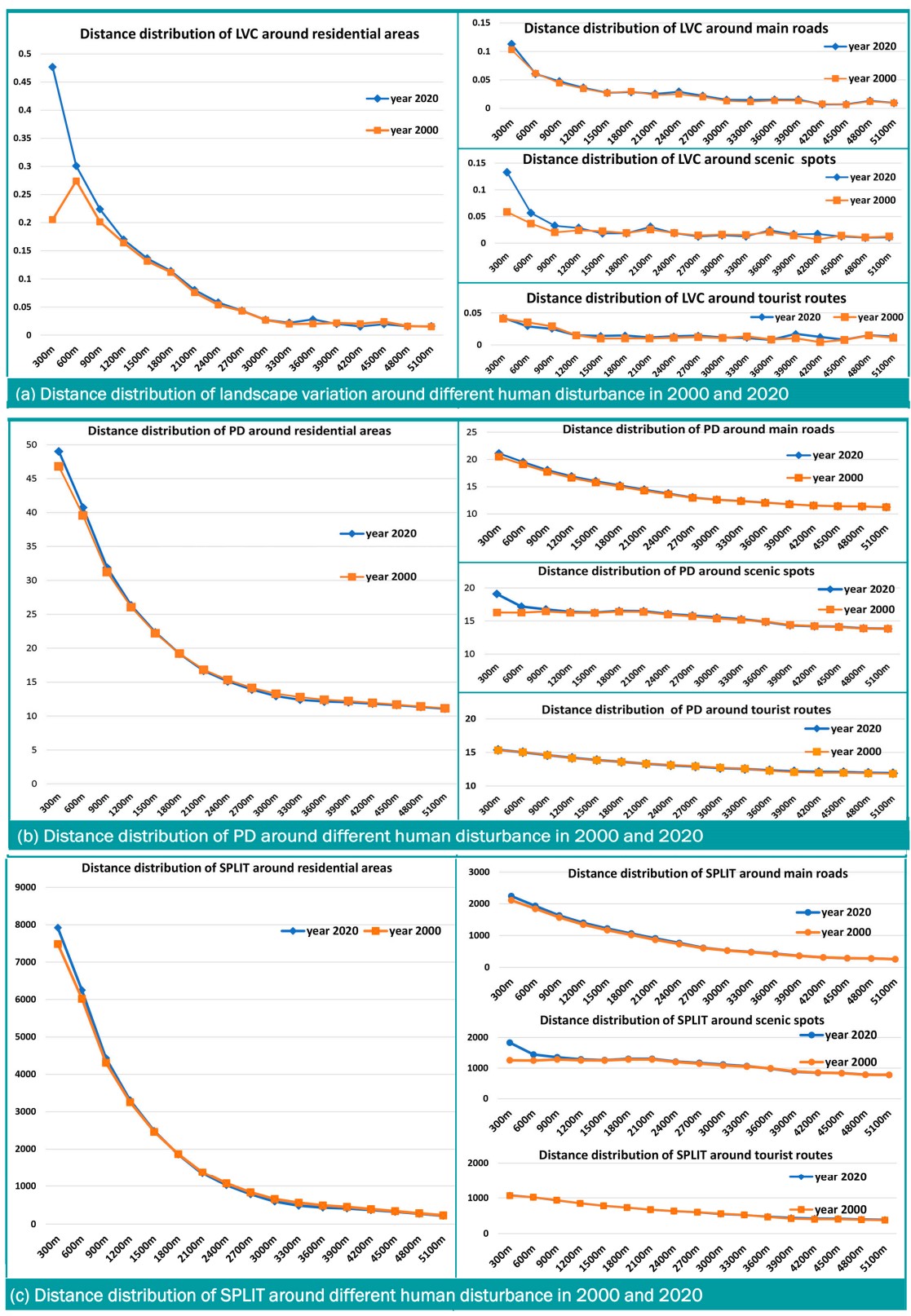

**Figure 12.** The distance distributions of the landscape fragmentation index around different human disturbances in 2000 and 2020. (**a**) Distance distributions of the LVC around different human disturbances in 2000 and 2020; (**b**) Distance distributions of the PD around different human disturbances in 2000 and 2020; (**c**) Distance distributions of the SPLIT around different human disturbances in 2000 and 2020.

## 5. Discussion

### 5.1. Spatial Heterogeneity of the Landscape Fragmentation in QLMNP

Regional differences in the landscape fragmentation of natural habitats exist in QLMNP. The loss of habitat integrity was found to occur mainly in the areas close to the boundaries and the edges of the protected area in QLMNP, while the fragmentation level is much higher in core protected zone in QLMNP. This result is in correspondence with previous research in other protected areas, they also found increasing fragmentation level in surround areas of protected polygons [12,34]. High value of landscape around national park may cause the increasing rate and degree of landscape fragmentation [33]. Since landscape fragmentation close to the boundaries may significantly affect the connectivity between the protected areas [8], further comprehensive analysis and a detailed control plan are required for the area surrounding the protected area of QLMNP to control the intense fragmentation of the natural habitat caused by anthropogenic activities.

QLMNP spans two provinces, namely Qinghai and Gansu, which are respectively located on the southern and northern slopes of the Qilian Mountains. Regional differences in the degree of fragmentation of natural habitats exist between communities in the two provinces. More communities in Gansu Province have been identified as having a high level of fragmentation in the natural habitat. Based on the differences between the habitat fragmentation of the two provinces, appropriate conservation measures are needed.

In further analysis, according to method of this paper, we can find the areas with the most serious habitat fragmentation in Qilian Mountains National Park, and then find finer and more detailed LULC data and natural and human condition data to study the drivers of landscape fragmentation in these areas.

### 5.2. Impact of Human Disturbance on Landscape Fragmentation in QLMNP

Compared with natural factors (the terrain niche index, annual mean temperature, and annual mean precipitation), human disturbances have greater impacts on fragmentation of natural habitat. This result is similar to previous research in QLMNP, which found that human activities have played a major role in habitat loss and fragmentation, especially to shrubland and grassland [16].

Further, among the various types of human disturbances, road, residential areas, settlements, and expansion of cropland have greater impacts on the landscape fragmentation of the natural habitat in QLMNP, which is consistent with the findings of previous studies analyzing the impacts of human factors on land-cover change in QLMNP [16,33]. The expansion of cultivated land and the built-up area is the main cause of landscape fragmentation. By contrast, the tourist route and scenic spot data revealed a relatively smaller impact on the landscape fragmentation of the natural habitat in QLMNP. According to other studies investigating the impacts of sources of disturbance on landscape patterns, tourism has had a greater impact on the landscape patterns in other protected areas, such as the Li River Basin and Wuyi Mountain National Park [25].

The number of tourists in Qilianshan National Park is lower than that of most other parks in southern and eastern China. By the end of August 2021, the tourist trajectory data of Pudacuo National Park, Wuyishan National Park, and Shennongjia National Park on Fooooooot and other websites were much higher than those of QLMNP, which may indicate that tourism in QLMNP has not been fully developed. This finding may be attributed to the low population density in northwestern China, which comprises grasslands, deserts, and snow-covered plateaus. It has been the world of nomads since ancient times [68]. Therefore, the national park is relatively large with a sparse population, relatively weak economic development, and poor transportation facilities. Thus, tourism has not been fully developed in QLMNP due to low accessibility and isolation compared with developed urban areas.

In further analysis, the impact of various human factors needs to be further quantified based on multi-source data. We can use big data from various social media application and quantify the extent and intensity of human activity.

### 5.3. Spatial Heterogeneity of the Drivers of Landscape Fragmentation in QLMNP

Spatial heterogeneity is reflected not only in landscape fragmentation, but also in the impacts of different human disturbances on landscape patterns. As the national park in the northwestern region of China, QLMNP covers a wide area [44] with large differences in the human activity and natural resources of various communities around the park. Based on the analysis results of this study, the main factors affecting the degree of landscape fragmentation vary with the communities in QLMNP.

Tourism activities have a relatively large impact on the central area of the national park, such as communities in Qilian County of Qinghai Province, while the road density and built-up area may have a greater impact on the eastern areas than the western areas of the national park. This may be because some towns in the midwest of the QLMNP, such as Mole and Muli town, were developed based on coal mining [69].

Previous research only mentioned the impact on the environment through construction of facilities [6], our research used multiple buffer rings to further quantify the impact of human activities and some facilities on habitat fragmentation in the surrounding area. The highest values of fragmentation index are in the first or second buffer zone, which is closest to human disturbance. Additionally, point-shaped sources of human disturbance, such as settlements and scenic spots, have a larger impact in the range of 900 m of the surrounding area compared with linear interference. This difference may be attributed to the human activities in the Qilian Mountains, which are affected by topography and altitude, and are mainly concentrated in a few important scenic spots and mining towns.

In further analysis, geomorphological factors should also be taken into account for a further analysis, especially for QLMNP, a mountain protected area. The terrain in QLMNP is complex and diverse, and the spatial heterogeneity of landscape fragmentation may be caused by terrain factors.

### 5.4. Sustainable Development of Communities in QLMNP: A Future Perspective

The results have strong implications for the development of QLMNP, especially for national park community planning. QLMNP covers a wide area, and although dozens of conservation stations have been set up in the park, the management and protection of QLMNP are closely related to the socioeconomic development of each community.

According to the results obtained by GWR, the effects of various human activity indicators in the Qilian Mountains vary greatly from east to west and from north to south. Compared with other national parks, QLMNP has a small number of tourists due to strict protection and the low level of tourist facilities, especially in the western areas, where tourism development in the protected area has been prohibited.

However, the results also showed that some tourist tracks pass through the core protected areas of QLMNP, which is not conducive to ecological protection. Further, according to multiple-ring buffer analysis, compared with 2000, a significant increase occurred in the level of fluctuation of the landscape fragmentation indexes in the surrounding areas of scenic spots in 2020. The overall impact of scenic spots on landscape fragmentation was found to be relatively small. However, based on the comparison between 2000 and 2020, the incremental impact of scenic spots on the degree of landscape fragmentation has been greater than those of roads and tourist routes. In addition, the important scenic spots in the Qilian Mountains are mainly distributed in relatively undisturbed natural habitats, such as snow-capped mountains and glaciers. Therefore, with the future development of tourism and the construction of roads, it is necessary to prevent damage to the ecological environment of these places.

Tourist activities must be further regulated. It is suggested that the government re-plan the core protected area and strengthen protection to prohibit tourists from entering the western and core areas of the Qilian Mountains characterized by stronger ecological sensitivity and weaker environmental capacity. However, in the eastern portion of QLMNP, the density of a few local communities with good tourism potential can be decreased to create a tourism support function consistent with local natural and cultural customs.

Indigenous residents can be encouraged and supported to engage in various forms of tertiary industries [45], such as accommodation, transportation, and folk performances, to increase their income. For instance, residents can be offered forest ranger and public welfare positions.

Because QLMNP is located at the source of the Yellow River and the inland rivers of northwest China, ecological protection is the most important goal. Additionally, ecological protection is a top priority in China's national park management [53]. Therefore, local communities can be encouraged to participate in the ecological conservation of QLMNP [70]. Some researchers have proposed that the economic benefits generated by the construction of national parks should increasingly benefit indigenous residents and local communities [71], which can deepen their identification with national parks and contribute to shared benefits. Concession rights in the fields of environmental education, recreational services, and ecological experience in national parks should be given to indigenous residents, with a focus on targeted poverty alleviation [70]. The government must support community residents to engage in environmental education and the experience of national parks [47] to ensure a relatively stable economy while participating in the construction and management of national parks [53].

## 6. Conclusions

Human disturbance has been one of the key elements driving changes in landscape patterns. This study analyzed the spatial distributions of landscape fragmentation and human activities, after which multiple-ring buffer and GWR analyses were used to quantify the relationships between landscape fragmentation and human activities and facilities, as well as the level of impact of each factor on the landscape fragmentation in different communities. This approach can facilitate the development of a detailed control plan based on the landscape fragmentation and socioeconomic development of each community in QLMNP.

It is found that habitat fragmentation in QLMNP could be significantly correlated with human activities, especially in the edge of the national park. The habitat fragmentation level of the communities with high density of scenic spots is not high. This is because the scenic spots in QLMNP are mainly natural habitat with intact landscapes. Moreover, the closer the distance to large cities, the higher the levels of the landscape fragmentation. Based on the impacts of human activities on the landscape fragmentation of the natural habitat, comprehensive regulations and legislations based on the fragmentation level and human activity in each community can be formulated and implemented. Communities in the large core protected areas in the central region of QLMNP should be prioritized for ecological protection. In the residential areas located outside the general protected area, the expansion of the built-up area and cultivated land should be controlled, and visitors should be received in these areas. Moreover, settlements located in highly ecologically sensitive areas should be relocated.

In China, the master plan of the first batch of national parks has basically been prepared [53] and now the plan needs to be gradually refined and adjusted according to the conditions of each community. Understanding the relationship between habitat fragmentation and human activity in various communities is necessary to formulate detailed conservation policies in each community.

Many studies on the impact of human activities on protected areas are still insufficient, especially for the phenomenon that the impact of human activities on the landscape pattern will gradually decrease with the increase of the distance from the main activity area. And different human activities, such as settlements, scenic spots, and roads [20], have different attenuation characteristics of the impact on landscape fragmentation. Identifying the anthropogenic factors that have the strongest impact on landscape fragmentation and limiting this anthropogenic activity around protected areas is important for management of protected area and mitigation of habitat fragmentation.

This study was limited by a lack of available data. The tourist route data were only available at websites associated with outdoor activities, and data on only a portion of the total sample were obtained for this study. Similar total numbers of visitor trajectory data could not be accessed. Moreover, environmental pressure due to factors such as pollution generated by community development could not be adequately quantified in this study. In addition, there are some disadvantages of using GWR in this field. As GWR is a sample point-based technique, the variables associated with each community are assumed to be samples obtained at the mean value of pixels in the core of the community. In a mountainous area, the area and shape of the communities differ substantially, and even the centers of the communities are outside the range, suggesting the need for further studies to corroborate the results.

**Author Contributions:** Conceptualization, B.R. and G.W.; methodology, B.R.; software, B.R.; validation, B.R. and G.W.; formal analysis, B.R.; investigation, B.R. and W.B.; resources, B.R.; data curation, B.R.; writing—original draft preparation, B.R.; writing—review and editing, G.W., K.P., A.S., J.Y. and M.M.; visualization, B.R.; supervision, G.W.; project administration, G.W.; funding acquisition, G.W. All authors have read and agreed to the published version of the manuscript.

**Funding:** This research was funded by the APFNet-UBC National Park Research Project (APFNet 2017 Sp2-UBC), and the program of China Scholarships Council (CSC) (Grant No. 202006010074).

**Institutional Review Board Statement:** Not applicable.

**Informed Consent Statement:** Not applicable.

**Data Availability Statement:** Not applicable.

**Conflicts of Interest:** The authors declare no conflict of interest.

### Appendix A

This Appendix A contains explanatory regression analysis of the fragmentation index and independent variables and the results of GWR and OLS analysis. Tables A1–A4 are the results of explanatory regression analysis. Tables A5 and A6 are the comparisons of results of GWR and OLS analysis.

**Table A1.** The highest adjusted $R^2$ results of the explanatory regression analysis of the PD and four independent variables.

| Adj $R^2$ | AICc | JB | K (BP) | VIF | SA | Model |
|:---:|:---:|:---:|:---:|:---:|:---:|:---:|
| 0.91 | −72.92 | 0 | 0 | 13.88 | 0.08 | +Proportion of cultivated land *** <br> −Proportion of built-up area *** <br> +Road density *** <br> −Annual mean temperature *** |
| 0.91 | −72.32 | 0 | 0 | 20.66 | 0 | −Kernel density of scenic spots ** <br> +Proportion of cultivated land ** <br> −Proportion of the built-up area *** <br> +Road density *** |
| 0.91 | −71.94 | 0 | 0 | 20 | 0 | +Proportion of cultivated land ** <br> +Road density *** <br> −Nighttime light level * <br> −Population density *** |

Variable significance: * = 0.10; ** = 0.05; *** = 0.01.

**Table A2.** The highest adjusted R$^2$ results of the explanatory regression analysis of the SPLIT and four independent variables.

| Adj R$^2$ | AICc | JB | K (BP) | VIF | SA | Model |
|---|---|---|---|---|---|---|
| 0.99 | −379.07 | 0 | 0 | 32.18 | 0.57 | +Proportion of cultivated land * <br> −Proportion of built-up area ** <br> +Road density *** <br> +Nighttime light level *** |
| 0.99 | −375.01 | 0 | 0 | 29.28 | 0.72 | +Potential productivity of cropland ** <br> −Proportion of the built-up area ** <br> +Road density ** <br> +Nighttime light level *** |
| 0.99 | −373.48 | 0 | 0 | 24.01 | 0.75 | −Proportion of the built-up area ** <br> +Road density *** <br> +Average mean temperature <br> +Nighttime light level *** |

Variable significance: * = 0.10; ** = 0.05; *** = 0.01.

**Table A3.** The highest adjusted R$^2$ results of the explanatory regression with the PD and one independent variable.

| Adj R$^2$ | AICc | JB | K (BP) | VIF | SA | Model |
|---|---|---|---|---|---|---|
| 0.64 | 8.08 | 0 | 0 | 1 | 0 | +Road density *** |
| 0.53 | 23.96 | 0 | 0 | 1 | 0 | +Proportion of cultivated land *** |
| 0.48 | 30.42 | 0 | 0.12 | 1 | 0 | +Population density *** |
| 0.36 | −40.37 | 0 | 0.77 | 1 | 0 | +Proportion of the built-up area *** |

Variable significance: *** = 0.01.

**Table A4.** The highest adjusted R$^2$ results of the explanatory regression with the SPLIT and one independent variable.

| Adj R$^2$ | AICc | JB | K (BP) | VIF | SA | Model |
|---|---|---|---|---|---|---|
| 0.98 | −327.14 | 0 | 0.87 | 1 | 0.67 | +Nighttime light level *** |
| 0.92 | −238.85 | 0 | 0.03 | 1 | 0 | +Proportion of built-up area *** |
| 0.88 | −210.43 | 0 | 0 | 1 | 0 | +Road density *** |
| 0.78 | −174.02 | 0 | 0 | 1 | 0 | +Population density *** |

Variable significance: *** = 0.01.

**Table A5.** The results of GWR and OLS analysis including the PD and five5 selected independent variables.

| AICc | | Adjusted R$^2$ | |
|---|---|---|---|
| GWR | OLS | GWR | OLS |
| −132.5720 | −129.2241 | 0.8916 | 0.8558 |

**Table A6.** The results of GWR and OLS analysis including the SPLIT and five selected independent variables.

| AICc | | Adjusted R$^2$ | |
|---|---|---|---|
| GWR | OLS | GWR | OLS |
| −375.9478 | −366.5200 | 0.9920 | 0.9914 |

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
