# Peer review of "Impact of Human Disturbances on the Spatial Heterogeneity of Landscape Fragmentation in Qilian Mountain National Park, China"

_land, doi:10.3390/land11112087_

Round 1

Reviewer 1 Report

Dear authors,

congratulations for your work and current paper. There are just few minor issues that should be solved before publishing:

Please rives line 52, 67, 83, 191, 207, 253 - minor English or typo errors that need to be corrected.

Check citations in lines 133, 138, to be accordingly.

Line 161 - Based on a previous study, nine factors of human disturbance 161 and three factors of the natural background were selected as independent variables (Table1). - could you please clarify this sentence and reason why you have chosen those particular indicators for your study, more detailed?

Please combine Table 1 and 2 and leave one table, since both refer to the indicators used for your study, and they are so much alike.

The rest of the paper is very well structured, and the results are clearly presented, I would suggest that to mention under each figure the input data sources, and one last recommendation is to keep the conclusions section more concise.

Good luck!

Author Response

Thank you so much for your feedback. Below are my responses and revision to those comments:

Point 1: Please rives line 52, 67, 83, 191, 207, 253 - minor English or typo errors that need to be corrected.

52, 67, 83 191,207, 253

Response 1: Thanks for your reminder. I have revised errors.

Point 2: Check citations in lines 133, 138, to be accordingly.

Response 2: Thanks for your reminder. I have revised errors.

Point 3: Line 161 - Based on a previous study, nine factors of human disturbance 161 and three factors of the natural background were selected as independent variables (Table1). - could you please clarify this sentence and reason why you have chosen those particular indicators for your study, more detailed?

Response 3: Based on reality of Qilian Mountains and previous studies, we would like to choose human settlements and residential areas, transport and travel demand, human infrastructure, and agricultural development as possible drivers of habitat fragmentation.  Based on previous study, hydropower stations and mining sites are also important factors that may have influence on landscape fragmentation and natural environment. But because these data cannot be obtained, we only select the indicators in Table 1 for analysis. And I added a few sentences to the paragraph above Table 1 to clarify reason why you have chosen those indicators.

Point 4: Please combine Table 1 and 2 and leave one table, since both refer to the indicators used for your study, and they are so much alike.

Response 4: Thanks for your suggestion. And sorry for causing misunderstanding. I think that I didn't write the header of the form clearly, which cause misunderstanding: Table 1 is mainly about references using indicators of the human impact on the landscape pattern, while Table 2 is mainly about the data source and calculation method of each indicator. And I have tried to combine Table 1 and Table 2 together. But due to the limitation of page width, the two tables could not fit, especially there are some equations in Table 2, which makes typesetting harder, so I still had to split them into two tables.

Point 5: The rest of the paper is very well structured, and the results are clearly presented, I would suggest that to mention under each figure the input data sources,

Response 5: Thanks for your suggestion. All the data source and calculation method are in Table 2.

Point 6: One last recommendation is to keep the conclusions section more concise.

Response 6: Thanks for your suggestions. I revised the Conclusion part to remove some irrelevant and redundant terms and phrases to make the whole paragraph more concise.

Reviewer 2 Report

Dear all, 

thanks for the invitation to review this interesting manuscript. In fact, I do believe the work has potential as well as scientific soundness.

However, I recommend the author to improve the following:

- the introduction and literature review should be separated sections

- the discussion should consider the adding of more references of similar studies and researches to foster the debate on this specific topic and consequently move forward in the thematic literature

best,

Author Response

Thank you so much for your feedback. Below are my responses and revision to those comments:

Point 1: the introduction and literature review should be separated sections

Response 1: Thanks for your suggestions. I have separated the sections of introduction and literature review.

Point 2: the discussion should consider the adding of more references of similar studies and research to foster the debate on this specific topic and consequently move forward in the thematic literature

Response 2: Thanks for your suggestions. And I agree with your suggestions. At this time, I added as many references of similar studies in the discussion section as possible for comparison.

But it is difficult to foster debate on this specific topic. The reasons are mainly due to two aspects: there are currently only a few literatures on human-induced influences on natural habitat in Qilian Mountains National Park, and the content of our research and the data used research are different. Therefore, it is difficult to find relevant specific research contents for comparison. Some anthropogenic drivers that may have a greater impact on the habitat fragmentation, such as mining sites and hydropower stations, are temporarily unavailable to us, so it is difficult to compare effectively with other literature in many aspects, and thus, it is difficult to foster the debate.

However, I have read and added as much related literature as possible, and I have rewritten sections 5.2 and 5.3 of the Discussion based on your suggestions. I compared the findings of this study with other related research findings, identified the inadequacies of my research, and made recommendations for future research. Among them, the conclusions of Section 5.1 are similar to those of previous studies on national parks.

Reviewer 3 Report

The subject of human influence on landscape fragmentation is very interesting, and of actuality. The conclusions mentioned in lines 22-23 are perfectly logical, since tourism is not the most disturbing human activity affecting the landscape.

Still, even if the authors chose not to include in the analysis the influence of the relief, I think it should be taken into account (for example lines 252-258). Landscape fragmentation can sometimes be caused by…landscape itself. So, so for a further analysis, geomorphological factors should be taken into account.

The paragraph between lines 45-53 should be revised. From the use of some terms to a decision if the activities mentioned could have an input or already have.

Line 58 – Be more precise, not all protected areas are surrounded by rural environments. If needed, make a more precise reference to the study area.

On the overall, a minor revision of English is needed (small examples lines 70-71, 76, 83).

In the overall analysis, did a time factor has been considered? The use of multi-temporal satellite images can be useful in deducing the evolution process of land fragmentation. The use of LCLU data for different periods might give an insight.

From my point of view, the most important conclusion is that from lines 371-377.

403-404 – language revision.

For a further study, I would take into account the density of accommodation facilities.

Author Response

Thank you so much for your feedback. Below are my responses and revision to those comments:

Point 1: Landscape fragmentation can sometimes be caused by…landscape itself. So, so for a further analysis, geomorphological factors should be taken into account.

Response 1: Yes, thanks for your suggestions. I agree with the point that geomorphological factors should be considered for a further analysis, especially in mountain protected areas. As elevation increases, the landscape changes. The degree of landscape fragmentation will also change accordingly. I will take more geomorphological factors in the future.  And I have added this points into 5.3 discussion section.

Point 2: Still, even if the authors chose not to include in the analysis the influence of the relief, I think it should be taken into account (for example lines 252-258).

Response 2: Thanks for your suggestions. And I have a question, does 'influence of the relief' mean that the influence of certain human factors is diminished in time and space? For spatial analysis, I used multiple buffer ring analysis to analyze landscape fragmentation around human disturbance in the following text.

Point 3: The paragraph between lines 45-53 should be revised. From the use of some terms to a decision if the activities mentioned could have an input or already have.

Response 3: Thanks for your suggestion. I made some changes to remove some irrelevant and redundant terms and phrases to make the whole paragraph more concise. But I don't understand the meaning of your sentence. Does it mean that the indicators used in my analysis later must include the terms mentioned in this paragraph.

Point 4: Line 58 – Be more precise, not all protected areas are surrounded by rural environments. If needed, make a more precise reference to the study area.

Response 4: Yes, I agree with your suggestion. Only some protected areas are surrounded by rural environments. I changed the way of expression and added some literature to make the expression more precise.

Point 5: On the overall, a minor revision of English is needed (small examples lines 70-71, 76, 83).

Response 5: Thanks for your suggestions. I have revised.

Point 6: In the overall analysis, did a time factor has been considered? The use of multi-temporal satellite images can be useful in deducing the evolution process of land fragmentation. The use of LCLU data for different periods might give an insight.

Response 6: Thanks for your suggestions. I really agree with your point that use of multi-temporal satellite images can be useful in deducing the evolution process of land fragmentation. In fact, the data that I used to calculate landscape metrics is GlobeLand30's products, mainly including 2000, 2010 and 2020 data. This data product is mainly the processed data obtained by synthesizing the images of all months of the year.

When I used the GWR model to analyze the impact of human factors on landscape fragmentation, due to data deficiencies, I only input data in 2020 into GWR model. As for some factors, such as visitor pressure (trajectories of visitors), I could not find data for 2000 and 2010.

However, when using multiple ring buffer analysis method to analyze the variations of the landscape fragmentation indexes in adjacent buffer zones under different types of human disturbance, the landscape metrics calculated by LULC of different year in 2000 and 2020 were compared.

Point 7: From my point of view, the most important conclusion is that from lines 371-377.

Response 7: Thanks for your suggestions. I have added this point to the conclusion section at the end of the paper.

Point 8: 403-404 – language revision.

Response 8: Thanks for your suggestions. I have revised.
